# Fungal Hal3 (and Its Close Relative Cab3) as Moonlighting Proteins

**DOI:** 10.3390/jof8101066

**Published:** 2022-10-11

**Authors:** Antonio Casamayor, Joaquín Ariño

**Affiliations:** Departament de Bioquímica i Biologia Molecular, Institut de Biotecnologia i Biomedicina, Universitat Autònoma de Barcelona, Cerdanyola del Vallès, 08193 Barcelona, Spain

**Keywords:** protein phosphatase, Ppz1, Vhs3, CoA biosynthesis, heterotrimer

## Abstract

Hal3 (Sis2) is a yeast protein that was initially identified as a regulatory subunit of the *Saccharomyces cerevisiae* Ser/Thr protein phosphatase Ppz1. A few years later, it was shown to participate in the formation of an atypical heterotrimeric phosphopantothenoylcysteine decarboxylase (PPCDC) enzyme, thus catalyzing a key reaction in the pathway leading to Coenzyme A biosynthesis. Therefore, Hal3 was defined as a moonlighting protein. The structure of Hal3 in some fungi is made of a conserved core, similar to bacterial or mammalian PPCDCs; meanwhile, in others, the gene encodes a larger protein with N- and C-terminal extensions. In this work, we describe how Hal3 (and its close relative Cab3) participates in these disparate functions and we review recent findings that could make it possible to predict which of these two proteins will show moonlighting properties in fungi.

## 1. Introduction

Moonlighting proteins can be defined as a group of multifunctional proteins in which a single polypeptide performs diverse biologically relevant functions, providing this protein is not the result of gene fusions or multiple RNA-splicing events. These kinds of proteins are found in mammals, bacteria and even viruses [1]. At the time of writing, the MoonProt database (http://www.moonlightingproteins.org/, accessed on 15 September 2022) has collected over 500 entries, of which around 60 correspond to fungal organisms. We present here the interesting case of Hal3, a yeast protein that was initially identified as a regulatory subunit of Ppz1, a fungal-specific Ser/Thr protein phosphatase and, subsequently, shown to participate in the biosynthetic pathway of the ubiquitous coenzyme A molecule.

From the point of view of the scientific literature, the gene *SIS2* (also known as *HAL3*) emerged in 1995, when two independent laboratories working with *S. cerevisiae* reported apparently unrelated findings concerning this gene. The group of Kim Arndt, in Cold Spring Harbor (USA), reported the isolation of a gene able, in high-copy numbers, to substantially increase the growth rate of a strain defective in the Sit4 protein phosphatase [2]. The gene was named *SIS2* (for the *sit4* suppressor) and encoded a protein of 562 amino acids, whose most striking feature was a long stretch extremely rich in acidic residues at its C-terminus. The beneficial effect of the overexpression of *SIS2* was due, at least in part, to the fact that it stimulated the rate of *SW14*, *CLN1* and *CLN2* mRNA accumulation, which is defective in *sit4* mutants. Shortly after, Ramon Serrano’s laboratory in Valencia (Spain), in collaboration with Gerald Fink (Cambridge, MA, USA), reported the isolation of the same gene as able to improve the growth of wild-type cells exposed to toxic concentrations of sodium and lithium ions, when the gene is overexpressed [3]. Hence, it was named *HAL3* for halotolerance (and we will use this denomination from now on). The deletion of *HAL3* yielded cells hypersensitive to high salt concentrations, and it was proposed that Hal3 could participate in the activation of potassium uptake and the efflux of sodium. In the latter case, this could occur by increasing the expression of the *ENA1/PMR2A* gene, encoding a P-type ATPase pivotal for sodium efflux under salt stress. One year later, Ramon Serrano’s group isolated and characterized a homolog of Hal3 in *Candida tropicalis* (CtHal3) and found that it partially complemented the salt sensitivity of a *S. cerevisiae hal3* mutant [4]. In this same work, the authors isolated a gene from *S. cerevisiae* (*YKL088w*) that encoded a protein with a significant identity and with Hal3 at its C-terminal half, although its ability to complement the *hal3* mutant was only partial. The relevance of *YKL088w* in this story will be discussed later.

## 2. Identification of Hal3 as an Inhibitor of the Ser/Thr Protein Phosphatase Ppz1

In 1992, our laboratory reported the cloning and characterization of a gene named *PPZ1*, encoding a putative Ser/Thr protein phosphatase of 692 amino acids [5]. The Ppz1 protein exhibited two different halves, one being a C-terminal domain structurally related to type 1 phosphatases, and the other being an N-terminal moiety unrelated to sequences found in other protein phosphatases. Shortly afterwards, Ppz2, a *S. cerevisiae* paralog of Ppz1, was isolated as a multicopy suppressor of the *slt2 (mpk1)* and *pkc1* mutations (which make cells prone to lysis due to defects in cell-wall construction), and it was shown that a *ppz1*Δ *ppz2*Δ *slt2*Δ mutant displayed a lytic phenotype, even at normal temperatures [6]. Further work confirmed the relevance of Ppz1 and Ppz2 in the maintenance of cell-wall integrity [6,7]. Notably, the study of a *ppz1* deletion mutant suggested a major role for the phosphatase in salt tolerance because these cells are hypertolerant to sodium or lithium cations in a way that is, at least in part, caused by an increase in the expression levels of the *ENA1* ATPase gene [8]. Later on, it was proposed that this effect was mediated by the inhibition of the calcium-dependent phosphatase calcineurin [9]. These phenotypes, opposite to those observed for the deletion of *HAL3*, were the lead for the identification of Hal3 as a negative regulatory subunit of Ppz1. Indeed, it was demonstrated that Hal3 could bind to the catalytic C-terminal domain of Ppz1 and inhibit its activity [10]. It was also shown that, in all cases, the disruption or overexpression of both genes provided opposite effects. Thus, the toxic effect caused by the overexpression of Ppz1 [11] was counteracted by the overexpression of Hal3, which also aggravated the lytic phenotype of an *slt2* mutant, thus mimicking the deletion of *PPZ1* in this genetic background [10]. Later on, it was shown that the effect on potassium influx observed in cells overexpressing Hal3 [3] was also mediated through Ppz1 inhibition of the high-affinity potassium transport Trk1 and Trk2 system [12]. In fact, cells lacking *ppz1* and *ppz2* showed an increased potassium uptake, leading to augmented intracellular turgor, and such an effect on potassium influx likely explains the impact of Ppz1 on the cell wall integrity pathway [13].

Further evidence of the role of Hal3 as an inhibitor of the Ppz1 function came from the identification of a change in the phosphorylation state of Tef5 (the transcription elongation factor EF1Bα) in ^32^P-labeled *ppz1 ppz2* cells in vivo, which was reproduced by an overexpression of Hal3 in wild-type cells [14]. The inhibitory activity of Hal3 on Ppz1 was also revealed by examining the effects of Hal3 overexpression in the *sit4* mutant, which were found to be entirely mediated by the Ppz1 phosphatase [15]. Just as the overexpression of Hal3 improves the growth of an *sit4* mutant, the combination of both mutations yields a synthetically lethal phenotype [2]. This phenotype was exploited in our laboratory to perform a high-copy suppressor screen using a conditional *sit4 hal3* mutant strain [16]. Such a screen yielded a number of genes, among which was *YOR054c*, renamed as *VHS3* (for viable in a *hal3 sit4* background), which is a gene-encoding protein with a substantial identity of Hal3 (about 40%), and is also endowed with an acidic C-terminal tail. Further work performed with the *vhs3* mutant and overexpressing strains showed that Vhs3 displays functions similar to those of Hal3, and opposing those of Ppz1 [17]. In addition, and similarly to Hal3, Vhs3 binds in vivo to the C-terminal catalytic moiety of Ppz1 and inhibits its phosphatase activity in vitro. Thus, Vhs3 qualified as a second regulatory subunit of Ppz1. However, it was clear that the physiological role of Vhs3 as a Ppz1 inhibitor was less relevant than that of Hal3.

## 3. Hal3 (and Vhs3) as Moonlighting Proteins: Their Involvement in CoA Biosynthesis

In the 2004 paper mentioned above [17], Ruiz and coworkers reported an interesting finding: while the independent mutation of *HAL3* and *VHS3* yielded cells perfectly viable (except under specific stress conditions), the double mutation was synthetically lethal. As mentioned above, it was known that the overexpression of Ppz1 has deleterious effects, so the lethal phenotype of the double *hal3 vhs3* mutant was initially interpreted to be the result of an unleashed Ppz1 activity due to the absence of its physiological inhibitors. However, the same authors demonstrated that the lethal phenotype was maintained even in the absence of *PPZ1* and *PPZ2*. This was the first clue that Hal3 and Vhs3 might perform essential functions in *S. cerevisiae* independently of their role in regulating Ppz1.

The first piece of information that paved the way leading to the identification of Hal3 and Vhs3 as moonlighting proteins came from the characterization of two putative homologs of Hal3 (AtHal3a and AtHal3b) in the plant *Arabidopsis thaliana* [18]. These authors showed that the expression of AtHal3a in *hal3* yeast mutants partially complemented their LiCl sensitivity, suggesting some degree of functional conservation between both proteins. However, AtHal3a was considerably shorter than ScHal3 or ScVhs3 (only 209 residues), it aligned with the region spanning from residue 250 to 480 in ScHal3, and it lacked the acidic C-terminal tail (Figure 1). Remarkably, the sequence of AtHal3a was reminiscent of that of a widespread family of eukaryotic and bacterial flavoproteins and, in fact, these authors showed that bacterially expressed AtHal3a contained non-covalently bound FMN [18]. A first step in understanding the function of AtHal3 came from the resolution of its 3D structure, showing that it might be an homotrimer, and the proposal that the protein might display enzymatic activity, specifically catalyzing the α,β-dehydrogenation of a peptidyl cysteine moiety [19]. Shortly after, Kupke and coworkers [20] reported a key finding: that AtHal3a catalyzed the decarboxylation of (*R*)-4-phospho-*N*-pantothenoylcysteine (PPC) to 4-phosphopantetheine (PP), which corresponds to the third step in coenzyme A biosynthesis, thus acting as a phosphopantothenoylcysteine decarboxylase (PPCDC). This work and subsequent reports [21,22] revealed the catalytic mechanism for AtHal3a PPCDC activity (Figure 2). Briefly, the active sites are found at the interfaces of the protomers, conforming the homotrimeric structure, and the catalysis takes place in two steps [19,20,21]. In the first step, the FMN-dependent oxidation of PPC’s cysteine, leading to the spontaneous decarboxylation of the resulting thioaldehyde, requires AtHal3a His^90^, N^142^ and M^145^ (the two latter within a P*X*MN*XX*MW motif, strongly conserved in eukaryotic and bacterial homologs). The second step involves a reduction in this intermediate to 4-phosphopantetheine and the reoxidation of the FMNH_2_ cofactor, and this step requires the AtHal3a redox-active Cys175 residue.

In spite of the detailed knowledge about the mechanism of action of AtHal3 and of its bacterial Dfp homologs [23], the nature of the PPCDC enzyme in yeast was unknown. All three likely candidates for yeast PPCDC, namely Hal3, Vhs3 and Ykl088w, presented the conserved His (corresponding to His^90^ in AtHal3a). However, only Ykl088w contains the conserved Cys and a PAMNTFMY motif that almost perfectly matches the conserved P*X*MN*XX*MW sequence described above. This fact, together with the observation that *YKL088w* is an essential gene [24], led to the proposal that the *YKL088w*-encoded protein was the actual yeast PPCDC enzyme. This proposal was additionally supported by the observation that the overexpression of Hal3 or Vhs3 could not complement the lethality of the *ykl088w*Δ mutation, while this was rescued by the overexpression of the *E. coli* bifunctional *coaBC* gene, which encodes a PPCDC activity [25]. Based on these findings, the authors proposed renaming *YKL088W* to *CAB3*, a denomination that has been widely accepted and will be used in this work from now on.

However, the notion regarding the nature of the PPCDC enzyme in yeast suffered a new turn when Ruiz and coworkers questioned the basis of the synthetic lethality of the *hal3 vhs3* mutant [26]. The authors reported that Cab3, contrary to Hal3 or Vhs3, did not behave as a Ppz1 inhibitor in vivo or in vitro. They also observed that the expression of *CAB3* was unable to rescue the lethality of the *hal3 vhs3* mutant. However, a crucial clue was derived from the observation that the expression of AtHal3a (as well as the human and mouse homologs) in heterozygous diploid *ykl088w*, *hal3 vhs3* and even triple mutant strains made it possible to recover viable haploid cells carrying the indicated mutations. These observations strongly argued in favor of the essential Cab3 and Hal3/Vhs3 functions being involved in the PPCDC activity within the CoA biosynthetic pathway. A second key observation was that, while none of the bacterially expressed Hal3, Vhs3 or Cab3 proteins individually showed any in vitro PPCDC activity, combinations of Hal3/Cab3 and Hal3/Vhs3/ Cab3 were active. Importantly, the mutation of H378 in Hal3, or that of H459 in Vhs3 (both equivalents to His90 in AtHal3a), to Ala prevented the rescue of the synthetically lethal phenotype of the *hal3 vhs3* mutation [26,27]. In contrast, the expression of a Cab3 version mutated in its H391 allowed the survival of the *cab3* strain. This indicated that the conserved histidine is functionally relevant in Hal3 and Vhs3 but not in Cab3. As expected, Cys478 in Cab3 proved to be important because its mutation to Ser yielded a protein unable to rescue the *cab3* deletion. Considering the roles of these residues in the two steps of the PPCDC-catalyzed reaction described above, it was considered that Hal3 or Vhs3 is required to complete the first step, and Cab3 is required to carry out its second step.

All these observations could be integrated, assuming an unanticipated concept: that *S. cerevisiae* PPCDC was not a homotrimer, as described for the *A. thaliana* [19] or human [28] enzyme, but a heterotrimer. Indeed, a model was proposed in which yeast PPCDC consists of at least one molecule of Cab3 and at least one molecule of Hal3 or Vhs3 (or their combination). In this model, the heterotrimeric enzyme has a single functional active site, formed at the interface of a Cab3 protomer, which donates the catalytic Cys (and the conserved PXMNXXMW sequence) and a molecule of Hal3 (or Vhs3), providing the required His. This model received strong support from chemical cross-linking experiments, showing the formation in vitro of the foreseen Hal3, Vhs3 and Cab3 trimeric structures, as well as from the reconstruction of an active PPCDC enzyme by the appropriate combination of bacterially expressed monomers [26]. Therefore, while in *S. cerevisiae* the functional role of Cab3 seems restricted to its intervention in the PPCDC heterotrimer, the previously identified function of Hal3 and Vhs3 in the regulation of the Ppz1 phosphatase protein allows them to be considered as moonlighting proteins (Figure 3).

## 4. Structural Insights into the Moonlighting Functions of Hal3

As described above, *S. cerevisiae* Hal3 and Vhs3 proteins are made of a conserved domain, similar in sequence and size to the PPCDC proteins from other eukaryotes (denoted here as PD, for PPCDC domain), an N-terminal extension (NtD, for N-terminal domain), which is predicted to be highly disordered, and the acidic C-terminal tail (CtD). It was demonstrated that a fragment of Hal3 corresponding to the PD region was sufficient to contribute to the PPCDC function in vivo [29]. However, when this domain or the PD–CtD fragment were combined in vitro with Cab3, the mixtures exhibited low PPCDC activity compared to that produced with a full-length Hal3 or the NtD+PD fragment. This suggested that the NtD of Hal3 somehow participates in the formation of an active heterotrimeric complex with Cab3. 

Subsequent work [30] showed that the PDs of Hal3, Vhs3 and Cab3 contain the structural elements required for PPCDC activity and trimerization. In contrast, these authors reported that Hal3 binds Ppz1 as a monomer, with a 1:1 stoichiometry. This implies that Hal3 must de-oligomerize from its usual homo- and heterotrimeric states. In addition, it was reported that Vhs3 and Hal3 PDs differ from each other on their secondary structure and mode of flavin binding. The data indicated that Vhs3 PD was more resistant to de-oligomerization compared with the Hal3 PD complexes, likely allowing Vhs3 PD to preferentially exist as a trimer, while Hal3 PD would be able to interchange between monomeric and higher-order oligomeric forms. In addition, it was shown that, while Hal3 PD can easily exchange from complexes containing only Hal3 PD and Cab3 PD, the addition of Vhs3 PD makes the heterotrimeric Hal3PD/Vhs3PD/Cab3PD complex much more stable and less likely to show a dynamic monomer exchange. These results were interpreted as Hal3 being more likely to act as Ppz1 inhibitor (in addition to participating in the formation of a functional PPCDC), while Vhs3 would mainly be involved in the stabilization of the heterotrimeric Hal3/Vhs3/Cab3 complex. This scenario would fit with previous observations deduced from the phenotypic analysis of *HAL3* and *VHS3* deletion or overexpressing strains [17,27], showing that, in vivo, Hal3 is the predominant regulator of Ppz1-mediated functions. In addition, according to existing datasets [31,32], the amount of cellular Hal3 exceeds that of Vhs3 from three- to six-fold. This difference is compatible with Hal3’s performance of dual roles as a Ppz1 inhibitor and as constituent of the *S. cerevisiae* PPCDC enzyme, while Vhs3 mainly performs as a component of the latter.

Upon examination of the 3D structure of the AtHal3 trimer and of the models created for the PD domain of ScHal3, Santolaria and coworkers postulated that residues G115 and L117 in AtHal3, and the corresponding L403 and L405 in ScHal3, could be possible components of a hydrophobic core important for generating and/or maintaining the trimeric state [33]. They found that the mutation of AtHal3′s G115 to Asn and that of L117 to Glu (or a combination of both) did not affect trimerization nor PPCDC function. When the changes L403N and L405E were introduced in ScHal3, the former had no effect, while the latter resulted in an inability to form homotrimers able to bind FMN. However, the L405E variant was still able to substitute for native ScHal3 in vivo, leading to the generation of functional PPCDC enzymes. This was interpreted as the L405E variant retaining the ability to efficiently interact with Cab3 and reconstitute a functional FMN-containing catalytic site. Remarkably, the L405E mutation decreased Hal3′s ability to interact with and to inhibit Ppz1 [33]. On the other hand, Olzhausen and coworkers [34] reported that, in addition to forming a trimeric structure, yeast Hal3, Vhs3 and Cab3 associate with other enzymes of the CoA-synthesizing pathway (Cab2, Cab4 and Cab5, but not with Cab1/PanK) to generate a large protein complex (CoA-SPC). The authors also postulated that Cab3 acts as a scaffold for the complex, and that the Cab3 N-terminal domain would be necessary for interaction with Cab2 and Cab5. 

In spite of the considerable amount of work done, it is still not fully known how Ppz1 is inhibited by Hal3 in *S. cerevisiae*. Ppz1 belongs to the type 1 phosphatase family, which is characterized by a catalytic subunit (PP1c) that interacts with and is controlled by a large number of regulatory subunits [35,36]. In the case of yeast PP1c (Glc7), many of its regulatory subunits contain an RVxF-consensus PP1c-binding motif (conserved in other eukaryotes), which binds to the Glc7 hydrophobic groove. Such a groove is strongly conserved and likely functional in Ppz1 [37]. However, the structural determinants for interaction between Ppz1 and Hal3 should differ substantially from those used by Glc7-regulatory subunits to bind to yeast PP1c because Hal3 does not bind Glc7 in vitro [10,38]. Remarkably, a putative Glc7-binding-like sequence (^263^KLHVLF^268^) can be found in Hal3 and is also present in Vhs3. Nevertheless, the mutation of Hal3′s H265 or F268 did not affect the binding or inhibitory capacity of Ppz1 [27], indicating that this motif is not functional for Ppz1 regulation. In this same work, most of the PD region of Hal3 was subjected to random mutagenesis, followed by functional screening. This approach yielded nine residues important for Ppz1-related Hal3 functions, of which seven clustered in a relatively short region, spanning from residues 446 to 480. Three of these mutations, V390G, I446K and W452G, decreased binding to Ppz1 and, as expected, were largely unable to inhibit Ppz1 in vitro. However, some mutations exhibiting normal Ppz1 interactions were totally (E460G) or partially (V462A) ineffective when tested for Ppz1 inhibition in vitro. This suggested that the Hal3 structural determinants required for Ppz1 binding and inhibition can be independent. In addition, it was found that the mutation of the conserved H378 to Ala (which, as described above, is deleterious for PPCDC function) did not impair Hal3’s inhibitory capacity or binding to Ppz1. This represented the first evidence that the structural Hal3 determinants relevant for Ppz1 regulation were different from those required for its role in the PPCDC enzyme.

It is worth noting that more recent experiments based on a random mutagenesis approach of the catalytic domain of Ppz1 [39] also yielded several mutations that rendered the phosphatase clearly refractory to Hal3 inhibition, without exhibiting a significant reduction in Hal3 binding. This work also strengthened the notion that the Hal3 ^263^KLHVLF^268^ motif was irrelevant for interaction with Ppz1. Because the most relevant Ppz1 mutations mapped to a conserved α-helix region used by mammalian inhibitor-2 to regulate PP1c, the authors postulated that the inhibitory strategy of Hal3 on Ppz1 might mimic that of inhibitor-2, even if both proteins exhibit almost no sequence similarity.

The earlier descriptions of Sis2/Hal3 already hinted about the functional relevance of the C-terminal acidic tail of this protein [2,3], even before it was known that these functions were mediated by Ppz1. The level of expression of a ScHal3 devoid of the C-terminal tail is normal: even so, it cannot mimic the functions of the entire protein that rely on Ppz1 regulation [29]. Although the precise role of this acidic stretch is unknown, it is worth noting that, when the C-terminal tail of the CnHal3b protein from *Cryptococcus neoformans*, which is not acidic, was replaced by the acidic tail of ScHal3, the hybrid protein gained the ability confer tolerance to lithium ions when expressed in *S. cerevisiae*, indicating that it was able to inhibit Ppz1 [40].

Hal3 can be recovered bound to Ppz1 when both proteins are simultaneously expressed in *E. coli* [39]. This fact was exploited to carry out chemical cross-linking experiments, involving full Hal3 and the C-terminal catalytic domain of Ppz1, followed by the identification of the covalently bound peptides by mass spectrometry [41]. It was found that most Hal3 links were restricted to the N-terminal half and the first third of the conserved PD domain. This was interesting because, although originally found unable to interact by itself with Ppz1 in vitro, the N-terminal extension of Hal3 was identified as relevant for its inhibitory function in vivo [29]. These observations were interpreted as if the initial interactions with Ppz1 may involve residues located within the PD region of Hal3, to be later reinforced by additional interactions with Hal3′s N-terminal section. Very recent work has revealed the existence of a short sequence in the N-terminus of *S. cerevisiae* Hal3 (^90^KRVPAVTFS^98^) that is crucial for the regulation of Ppz1 function and activity [42]. A slight variation of this sequence was also found in Vhs3 (^86^KRIPTVTFS^94^) but not in Cab3. Mutations in this sequence, mainly those affecting residues K90, R91, V95 and F97, are particularly detrimental to the function of Hal3 as an inhibitor of Ppz1. Interestingly, these mutations do not decrease the interaction between Hal3 and the phosphatase at all, indicating that they are important for the regulation of activity but not for binding. This fits with the earlier observation that Cab3 interacts with Ppz1 but does not inhibit the phosphatase [26]. This sequence is conserved in a subset of fungi and, as will be discussed later, it could serve to estimate the relevance of Hal3 or Cab3 proteins in regulating fungal Ppz enzymes.

As mentioned above, an overexpression of Ppz1 is very toxic for the cell, leading to a strong growth arrest [10,11,43]. This effect results from an excess of phosphatase activity [44] and likely derives from the alteration of multiple targets [45,46,47]. In any case, it was known for a long time that an overexpression of Hal3 was able to fully counteract the toxic effects of the excess of Ppz1 [10,44]. Ppz1 is normally found at the cell periphery, likely attached to the plasma membrane because of its conserved myristoylated Gly2, and the beneficial effect of Hal3 overexpression was thought to derive from the blocking of the phosphatase activity at the normal Ppz1 cellular location. However, recent work by Albacar and coworkers [48] showed that, in Ppz1-overexpressing cells, Hal3 recruits the phosphatase to internal membranes (mostly vacuolar) and that such intracellular trafficking is crucial to normalize growth. Interestingly, removal of the Hal3′s ^90^KRVPAVTFS^98^ motif described above also affects, even moderately, both Ppz1 intracellular relocalization and counteraction of toxicity in cells overexpressing the phosphatase.

## 5. The Moonlighting Capacity of Hal3 (and Cab3) across Fungi

Genes encoding Ppz-like phosphatases can be found in all fungi and, in all cases, the structure of the protein consists of a conserved C-terminal catalytic domain and a less-conserved N-terminal extension of a variable size. The enzyme has been characterized in diverse fungi, such as *Schizosaccharomyces pombe* [49] *Neurospora crassa* [50], *Candida albicans* [51,52], *Debaryomyces hansenii* [53], *Aspergillus nidulans* [54], *Cryptococcus neoformans* [40] and *Ustilago maydis* [55]. In most cases, in these organisms the Ppz phosphatase appears involved in the maintenance of monovalent cation homeostasis. Remarkably, in all cases tested, the recombinant phosphatase was effectively inhibited in vitro by *S. cerevisiae* Hal3 [40,55,56].

In contrast, the comparison of Hal3 and Cab3-like fungal proteins led to various scenarios (Figure 4). Virtually all members of the Saccharomycotina subphylum encode one protein with the structural characteristics of Cab3, and at least one Hal3/Vhs3 polypeptide (or both, in those suffering the whole genome duplication event). In these organisms, it is expected that the PPCDC enzyme will be a heteromer composed of at least two different Cab3 and Hal3/Vhs3 molecules. Almost without exception, these proteins are endowed with a relatively long N-terminal extension (>150 residues). Among them, most species of Saccharomycetaceae, except some organisms belonging to the genus *Kazachstania*, *Tetrapisispora*, *Vanderwaltozyma* and *Naumovozyma*, maintain the consensus sequence in the N-terminal extension of Hal3 mentioned above, at least in the form BBxxxVTF (where “B” implies a basic residue, K or R) (Table 1). The same pattern can be found in the Saccharomycodaceae family, and in at least one member of the Pichiaceae family (*Candida boidinii*). Remarkably, the signature seems absent in the Hal3 proteins of the Phaffomycetaceae and Dipodascaceae families, as well as in organisms classified within the CUG-Ser1 and the Saccharomycetales *incertae sedis* clades. In these cases, however, closely related sequences, in the form of BxxxxVSF, xBxxxVSF or highly conserved variants, are detected in the N-terminal extension of the Cab3 protein.

In this regard, it is worth noting that in the case of *C. albicans,* which belongs to the CUG-Ser1 clade, it was shown that it was Cab3, and not Hal3, the protein acting as a Ppz1 inhibitor [57]. More recent work has demonstrated that, when the CaCab3 N-terminal region containing the relevant sequence (VRNQSVSFS) is replaced by the corresponding one in CaHal3, the resulting protein no longer acts as a Ppz1 regulator [42], suggesting that, in *C. albicans*, the moonlighting role is assumed by Cab3 and not by Hal3. As indicated in Table 1, nearly all members of the Saccharomycotina subphylum exhibit an acidic C-terminal tail. As mentioned above, this is important for the regulatory role of Ppz1. Based on these data, it is reasonable to assume that, in the Saccharomycotina subphylum, either Hal3 or Cab3 are moonlighting proteins.

Contrary to this situation, the Taphrinomycotina *S. pombe,* a few other ascomycetes, and all Basidiomycetes, almost without exception, have a single putative PPCDC-encoding gene, which contains all required structural elements for catalytic activity. In some cases, such as *Neurospora crassa* and *Cryptococcus neoformans*, two related genes can be found and, at least in the case of *C. neoformans*, both are known to be functional PPCDCs [40]. With a few exceptions (such as in *Ustilago maydis*), the PPCDC proteins in these species lack the N-terminal extension and, when it is present, they do not exhibit the BBxxxVTF sequence or any related variant. Furthermore, although in a few cases they are endowed with a C-terminal extension, this is not unusually acidic. All these structural characteristics suggest that, in this species, the PPCDC protein would not be a Ppz1 regulator and, therefore, would not display moonlighting properties. Indeed, in all cases that these shorter versions have been tested as Ppz inhibitors, such as the proteins from *S. pombe* [56], *C. neoformans* [40] or *U. maydis* [55], they showed a very poor or null inhibitory capacity, although their respective Ppz phosphatases were potently inhibited by ScHal3. These findings reinforce the notion that the presence of an N-terminal extension in Hal3/Cab3-like proteins containing close variants of the specific sequence ^90^KRVPAVTFS^98^ found at the N-terminus of ScHal3 would be instrumental to defining their role as a Ppz1-regulatory subunit and, hence, their potential as moonlighting proteins.

## 6. Concluding Remarks

The perception that, in *S. cerevisiae*, Hal3 acts as a moonlighting protein is widely established, as exemplified by the fact that this protein is included in the MoonProt database (ID 255). This might lead to the impression that Hal3 would be a moonlighting protein in all fungi. In this review, we show that this is not the case. Outside the Saccharomycotina subphylum, this protein would act only as PPCDC enzyme (likely as a homotrimeric complex), being unable to function as a regulator of Ppz1 activity. Within the Saccharomycotina subphylum, PPCDC would form a heterotrimeric complex with Cab3, and, in most cases, the Hal3 component would act as a Ppz1 inhibitor, thus qualifying as a moonlighting protein. However, in the CUG-Ser1 clade and a few other families, the roles appear exchanged and Cab3 (but not Hal3) would both regulate Ppz1 and contribute to PPCDC function. The perception that Cab3 is a multifunctional protein in some fungi is not yet widespread. For instance, at the time of writing, the MoonProt database does not include *C. albicans* Cab3 as a moonlighting protein, in spite of the compelling evidence accumulated [42,57].

A major challenge for future studies would be to unequivocally dissect the structural determinants required for Hal3 or Cab3 to bind and inhibit fungal Ppz phosphatases, which is, in fact, the ability that confers to them the quality of moonlighting proteins. In spite of the advances made, even for *S. cerevisiae*, the picture of how Hal3 regulates Ppz1 is blurry and incomplete. This is due, in part, to the fact that attempts to crystalize and solve the tridimensional structure of the complex have been unsuccessful, despite the fact that full-length ScHal3 can be recovered bound to the C-terminal catalytic domain of Ppz1 when both proteins are co-expressed in *E. coli* [39].

## Figures and Tables

**Figure 1 jof-08-01066-f001:**
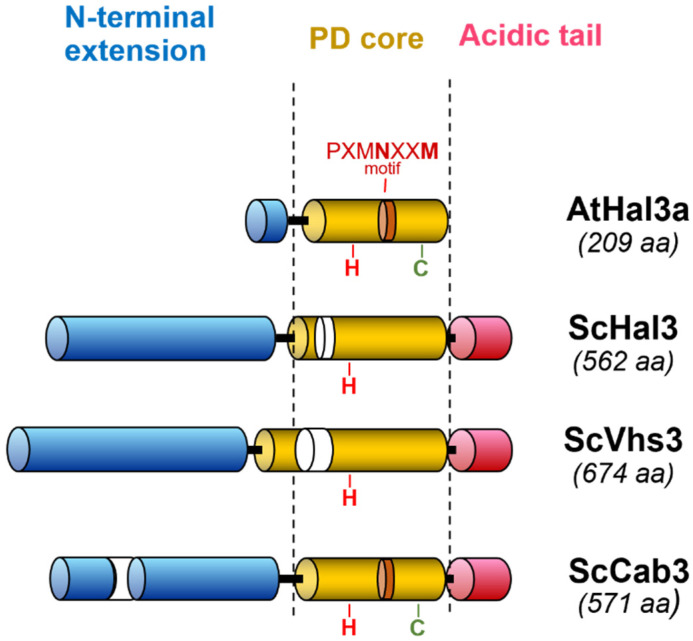
Comparison of overall *A. thaliana* AtHal3a with the *S. cerevisiae* Hal3, Vhs3 and Cab3 proteins. The core PPCDC domains are shown in gold, the N-terminal extensions in blue and the C-terminal acidic tails in red. Sequence insertions specific to the protein is denoted by uncolored segments. The location of the catalytic residues described in the text is shown.

**Figure 2 jof-08-01066-f002:**
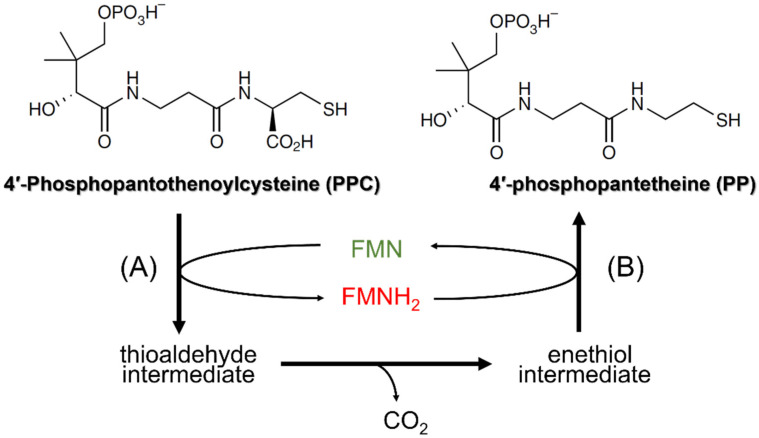
The reaction catalyzed by PPCDC. (**A**) denotes the flavin-dependent oxidation of PPC to the thioaldehyde intermediate, a reaction that requires the conserved histidine residue. After spontaneous decarboxylation of the intermediate, the conserved cysteine mediates the reduction of the enethiol to yield the final reaction product 4′-phosphopantetheine (**B**), regenerating the oxidized flavin.

**Figure 3 jof-08-01066-f003:**
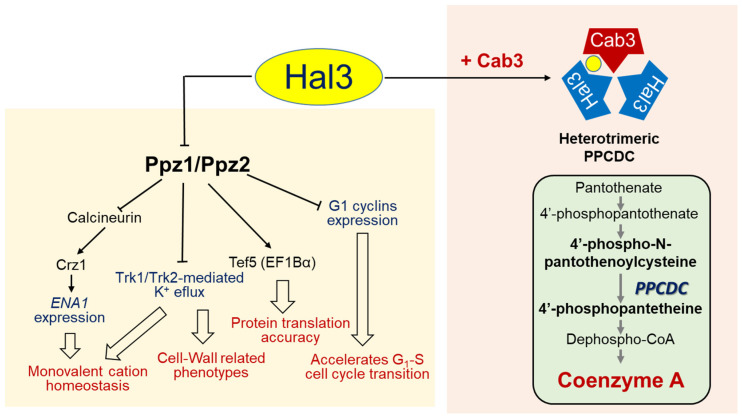
Cartoon depicting Ppz-dependent and independent functions of Hal3 in *S. cerevisiae*. See main text for details. The yellow circle on the right denotes the sole catalytic PPCDC site formed in the heteromeric PPCDC enzyme.

**Figure 4 jof-08-01066-f004:**
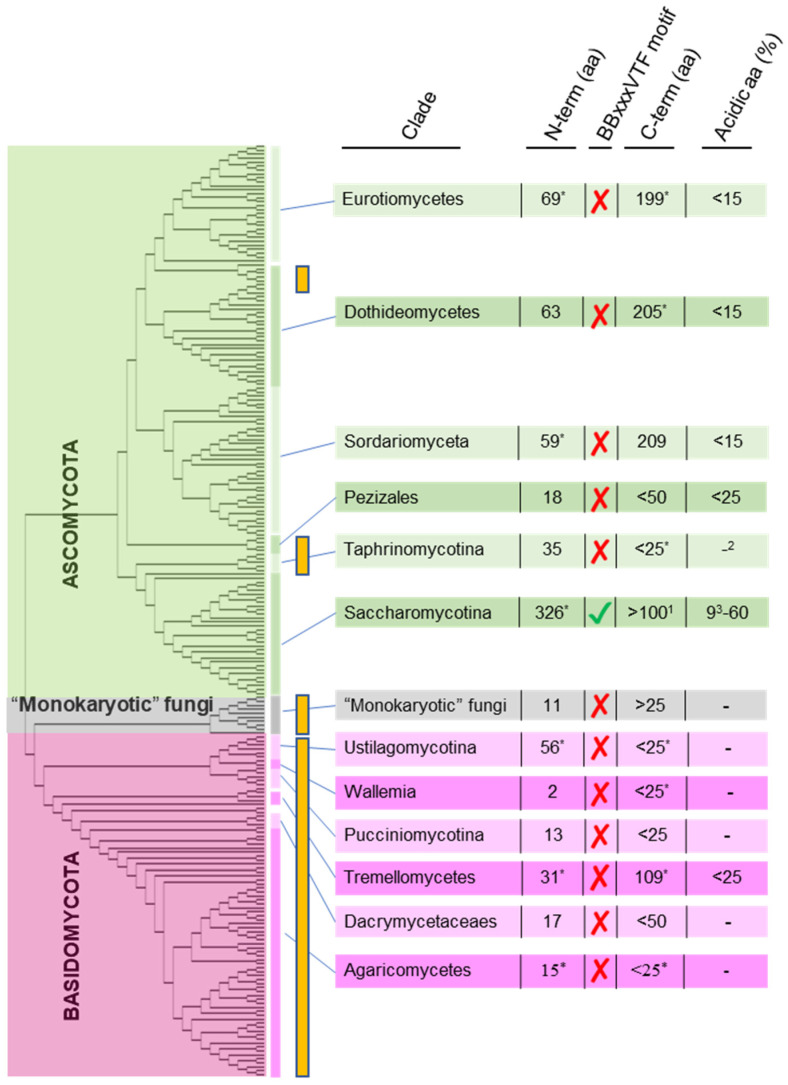
Structural features of Hal3/Cab3-like proteins in fungi. Phylogenetic tree of set of fungal species analyzed in [58]. The vertical orange bars denote clades with organisms with a monomeric PPCDC. The number of N- and C-terminal amino acids were calculated considering, as a central core, the cl19190 domain and excluding proteins in which other domains were identified in these extensions. The NCBI’s Batch Conserved Domain Search tool [59] was used to identify the location of the predicted domains in the output of the BLAST analysis for each specified family. Numbers in the figure indicate the average values. A threshold limit of at least 50 residues was required to calculate the percentage of acidic residues. (*), Clades including organisms with other domains identified in the N- or C-terminal regions (not considered for calculation). (1) Heterogeneous values ranging from nine to 480 residues due to the values obtained in a few species included in the Trichosporonaceae family (see Table 1); (2) in the genus *Schizosaccharomyces*, these values range between 43 and 60% when the residues included in the thymidylate synthase domain were not considered as a C-terminal extension. (3) The lowest range limit is due to the few members of the Trichosporonaceae family (see Table 1).

**Table 1 jof-08-01066-t001:** Structural features of the Hal3/Cab3 N- and C-terminal extensions in the Saccharomycotina subphylum. “B” denotes basic amino acids. Functionally relevant residues (as defined in [42]) are shown in red and specific variants in light blue. Calculations for size and compositions were conducted as shown in Figure 4.

Family	Mean N-Term (Hal3 & Cab3)	Consensus Sequence at the N-Terminal Extension		Acidic C-Tail (% aa)
		**Hal3**	**Cab3**	**Hal3/Cab3**

**Saccharomycetaceae**	356	KRxxxVTF ^(1)^	--xxxVSF ^(2)^	65.8/67.6
**Saccharomycodaceae**	578	BBxxxVTF	-	49.5/46.8
**Phaffomycetaceae**	208	-	K-xxxVSF	51.3/51.4
			KRxxxISF ^(3)^	
			--xxxVKF ^(4)^	
**Pichiaceae**	384	-	-	50.6/52.7
		RKxxxVTF ^(5)^		
**CUG-Ser1 clade**	252	-	--xxxVSF	46.1/48.0
			-RxxxVSF ^(6)^	
**Dipodascaceae**	173	-	-	39.5/39.7
			SRxxxVSF ^(7)^	
**Saccharomycetales inc. sed.**	180	-	K-xxxVSF ^(8)^	44.4/46.4
**Trichosporonaceae**	48	-	-	9.4

Notes: (1) Except some organisms of genus Kazachstania, Tetrapisispora, Vanderwaltozyma and Naumovozyma. (2) Except Kazachstania unispora. (3) Komagataella. (4) Cyberlindnera fabianii. (5): Candida boidinii. (6) Candida dubliniensis, Candida albicans, Candida viswanathii, Candida haemuloni and Candida auris. (7) Yarrowia. (8) Diutina rugosa.

## Data Availability

Not applicable.

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
