# Peer review of "Fungal Hal3 (and Its Close Relative Cab3) as Moonlighting Proteins"

_jof, 2022, doi:10.3390/jof8101066_

Round 1

Reviewer 1 Report

This review paper on “Fungal Hal3 (and Cab3) as moonlighting proteins” was submitted to the Special Issue: Multifunctional Fungal Proteins. It is well fitted to the aim of the special issue and the authors have given important contributions to the uncovering of the biochemistry of Hal3 related proteins and know quite well the history of the scientific development of the topic. Therefore, the article is scientifically sound and well written.

 As a reader, I have a few comments on the manuscript. I hope they may help the authors to improve the paper considering its interest to a broader audience.

i)                    The title is quite intriguing and after reading the manuscript it continues unclear to me in which concerns the reference to “Hal3 (and Cab3)”. May be “Hal3 and its close relative Cab3” would be more clear. However, I understand that the issue is that, apparently, Cab3 being a moonlighting protein cannot be generalized to all fungi/yeasts.

ii)                  In the MoonProt database (http://www.moonlightingproteins.org/), S. cerevisiae Hal3 is already included as a moonlighting protein (#255) and associated to the functions described in this review. This fact is ignored in the review paper. For this reason, the phrase “... will review recent findings that might allow PREDICTING moonlighting properties for Hal3 (and Cab3) fungal proteins.” does not appear to be appropriate.

iii)                Differently, Cab3, is not included in the MoonProt database. It would be important to discuss this issue as well as to render it more clear.

iv)                The focus of the review paper is on protein chemistry and molecular biology. I missed the more physiological aspects. In order to broadening the interested audience and the paper impact, I would recommend more emphasis on the multifunctional role of Hal3 and, if adequate of Cab3, in the fungal cell.

Author Response

We appreciate the positive opinion of the referee and thank him/her for the suggestions.

i) The title is quite intriguing and after reading the manuscript it continues unclear to me in which concerns the reference to “Hal3 (and Cab3)”. May be “Hal3 and its close relative Cab3” would be more clear. However, I understand that the issue is that, apparently, Cab3 being a moonlighting protein cannot be generalized to all fungi/yeasts.

R) We agree in that the title sounds intriguing and, it fact, this was our objective: to trigger the interest of the reader into the “mystery” of the moonlighting properties of Hal3 and Cab3. Indeed, the point is that Cab3 likely replaces Hal3 as moonlighting protein only in a few fungal clades. Certainly, incorporating the clarification suggested by the referee provides some additional relevant information, without losing its “enigmatic” quality. Therefore, we are proposing as title “Fungal Hal3 (and its close relative Cab3) as moonlighting proteins.

ii) In the MoonProt database (http://www.moonlightingproteins.org/), cerevisiae Hal3 is already included as a moonlighting protein (#255) and associated to the functions described in this review. This fact is ignored in the review paper. For this reason, the phrase “... will review recent findings that might allow PREDICTING moonlighting properties for Hal3 (and Cab3) fungal proteins.” does not appear to be appropriate.

iii)                Differently, Cab3, is not included in the MoonProt database. It would be important to discuss this issue as well as to render it more clear.

R) Combined to ii) and iii). We were aware, of course, that the MoonProt database includes S. cerevisiae Hal3. However, this entry refers exclusively to the Hal3 protein from this specific organism. A primary goal of our review is to make the reader aware that, in some fungi, it is Cab3 and not Hal3 that acquires moonlighting properties, and that the recent advances in understanding the structural requirements for Ppz1 negative regulation allows predicting which of these two proteins will act as a moonlighting one. We have rewritten the last paragraph of the abstract in an attempt to make this important point clearer.

It is also true that Cab3 is not included in this database, but this simply means that the database is not comprehensive enough. The moonlighting properties of Cab3 in C. albicans were clearly described in our Petrényi et al. 2016 paper (ref. 61), in which, in the last sentence of the abstract, we wrote ““The fact that only CaCab3 exhibits its phosphatase regulatory potential in vivo suggests that in C. albicans CaCab3, but not CaHal3, acts as a moonlighting protein.” Further evidence for this role was provided in our recent paper Santolaria et al 2022 (ref. 46). In any case, we have added at the end of the paper a new section (6. Concluding Remarks) in which we discuss the points raised here by the referee.

Reviewer 2 Report

The manuscript titled "Fungal Hal3 (and Cab3) as moonlighting proteins" is a substantially well-written summary of the current literature. Given the thorough, comprehensive information, it is highly recommended that the manuscript be containing more relevant diagrams or models to attract the reader. In addition, it is highly recommended that the manuscript be containing a concluding remark/future trends of research in the subject area. With these new components added to the current version, the revised manuscript will attract more readers and secure more citations in the future.

Author Response

We thank very much the referee for his/her encouraging comments on our work and for the recommendations. We feel that these are really worth to consider, so we have added a new figure (Figure 3), in order to clarify the diverse physiological roles of Hal3, and prepared a last section, entitled “Concluding Remarks” in which we summarize the main “take home messages” and discuss future challenges in this field.